# Photocurable Thiol–yne Alginate Hydrogels for Regenerative Medicine Purposes

**DOI:** 10.3390/polym14214709

**Published:** 2022-11-03

**Authors:** Michael Zanon, Laura Montalvillo-Jiménez, Paula Bosch, Raquel Cue-López, Enrique Martínez-Campos, Marco Sangermano, Annalisa Chiappone

**Affiliations:** 1Dipartimento di Scienza Applicata e Tecnologia, Politecnico di Torino, C.so Duca Degli Abruzzi 24, 10129 Turin, Italy; 2Departamento de Química Macromolecular Aplicada, Instituto de Ciencia y Tecnología de Polímeros, Consejo Superior de Investigaciones Científicas (CSIC), C/Juan de la Cierva 3, 28006 Madrid, Spain; 3Grupo de Síntesis Orgánica y Bioevaluación, Instituto Pluridisciplinar (UCM), Unidad Asociada al Instituto de Ciencia y Tecnología de Polímeros, Instituto de Química Médica (CSIC), Paseo de Juan XXIII 1, 28040 Madrid, Spain; 4Dipartimento di Scienze Chimiche e Geologiche, Università Degli Studi di Cagliari, Via Università 40, 09124 Cagliari, Italy

**Keywords:** alginate hydrogels, click chemistry, thiol–yne reactions, tissue engineering

## Abstract

Every year millions of people worldwide undergo surgical interventions, with the occurrence of mild or severe post-treatment consequences meaning that rehabilitation plays a key role in modern medicine. Considering the cases of burns and plastic surgery, the pressing need for new materials that can be used for wound patches or body fillers and are able to sustain tissue regeneration and promote cell adhesion and proliferation is clear. The challenges facing next-generation implant materials also include the need for improved structural properties for cellular organization and morphogenic guidance together with optimal mechanical, rheological, and topographical behavior. Herein, we propose for the first time a sodium alginate hydrogel obtained by a thiol–yne reaction, easily synthesized using carbodiimide chemistry in a two-step reaction. The hydrogels were formed in all cases within a few minutes of light irradiation, showing good self-standing properties under solicitation. The mechanical, rheological, topographical, and swelling properties of the gels were also tested and reported. Lastly, no cytotoxicity was detected among the hydrogels. Soluble extracts in culture media allowed cell proliferation, and no differences between samples were detected in terms of metabolic activity and DNA content. These results suggest the potential use of these cytocompatible hydrogels in tissue engineering and regenerative medicine.

## 1. Introduction

Hydrogels are three-dimensional hydrophilic polymer networks that are able to absorb and retain a large amount of water without dissolving or losing their characteristic resistance properties [1,2,3,4,5,6,7]. Since Wichterle and Lim provided this definition and proved their use in ophthalmology [8], their application has expanded to cover a wide range of fields such as agriculture [9,10], sensors [11,12], and water-treatment solutions [13,14,15]. Nevertheless, since the beginning, their main area of implementation has been in biomedical systems, and especially tissue engineering, due to their extreme analogy with the mammalian extracellular matrix (ECM) [4,5]. In fact, hydrogels are often designed to support cell proliferation, differentiation, and migration. To facilitate these mechanisms, the gels typically contain a percentage of polymer between 0.1 to 10% in weight, giving rise to macroscopic porous architectures. Moreover, the intrinsic structure permits oxygen and nutrient transport, providing cells with an optimal hydrated 3D environment that mimics the native soft tissue [16]. One of the main challenges in the sector is choosing the correct material for matrix construction; different approaches have been implemented in the past to accomplish the task, i.e., employing synthetic [17], natural [18] or hybrid polymer solutions [19]. Generally, natural polymers are chosen because of their enhanced properties that are not easily reproducible in the laboratory setting, such as their biocompatibility and biodegradability. Notably, their easy extraction from natural sources, even from waste, and their abundance also make them favorable materials, especially to meet current sustainability criteria. Alginate is a natural polysaccharide usually extracted from brown algae through alkali solutions (mostly NaOH treatments) [20]. It is formed by variable structural unit blocks of (1→4) linked β-D-mannuronic acid and α-L-guluronic acid, depending on the algae species and its original geographical area of production/growth [21]. Both structural units exhibit partially deprotonated carboxylic groups under biological conditions, allowing hydrogel formation by ionic crosslinking with divalent cations (e.g., Ba^2+^ and Ca^2+^). Nonetheless, sudden changes in the hydrogel pH gradually restore the -COOH moieties, leading to hydrogel dissolution [22,23]. Precisely for this reason, the alginates are often modified with functional groups and then chemically crosslinked using different approaches. As a simple and rapid technique, photopolymerization is widely used to produce chemically crosslinked hydrogels in tissue engineering, thanks to the mild reaction conditions of room-temperature physiological environments [24]. Acrylate- and methacrylate-functionalized natural polymers are some of the most frequently used materials for processing scaffolds in tissue engineering, even though it is commonly known that acrylate is toxic at medium/high concentrations [25,26]. Instead, “click-chemistry” reactions, which produce high yields under mild conditions (e.g., water-based environments or room temperature), are frequently recognized as rapid, versatile, and regiospecific systems [27,28,29]. The huge popularity of these materials combined with their increased biocompatibility make them highly attractive in tissue engineering [30,31,32]. Thiol–yne reactions exploit the spatiotemporal, orthogonal, and extremely selective reactions between alkyne and thiols, leading to more homogeneous networks compared to (meth)acrylates [33,34,35]. Within this framework, in this study, we successfully functionalized alginate using propargylamine in a two-step reaction to modify a polysaccharide with yne moieties. The further addition of two types of dithiol crosslinkers enabled the formation of the first reported alginate thiol–yne hydrogel in the literature, whether under stoichiometric conditions or not. The rheological, mechanical, and swelling properties of the hydrogels were evaluated to prove the simplicity of their formation as well as their microscopic porous structures. Lastly, the cytocompatibility of the materials was checked through cell proliferation analysis, including metabolic activity and DNA content assays.

## 2. Materials and Methods

### 2.1. Materials

Sodium alginate from brown algae (SA, low viscosity); propargylamine (PA, 98%); N-(3-Dimethylaminopropyl)-N′-ethylcarbodiimide hydrochloride (EDC, ≥98%); N-hydroxysuccinimide (NHS); 1,4-dithiothreitol (DTT); poly(ethylene glycol) dithiol (PEG-SH, average Mn 1000); lithium phenyl-2,4,6-trimethylbenzoylphosphinate (LAP, Z95%); and hydrochloric acid solution (37%) were all purchased from Sigma-Aldrich and used as received without further purification. Sodium hydroxide pellets were purchased from Panreac, and pre-wetted dialysis membranes (MWCO 3500 Da Spectra/Por6) from Spectrum Laboratories.

### 2.2. Synthesis of Alginate yne

A 1.5 wt% SA solution was prepared by dissolving 3 g of SA in 200 mL of DI water in a round-bottom flask. Drops of aqueous HCl were slowly added to the flask until reaching pH = 4. In a vial, the corresponding amounts of EDC and NHS, in a 1:1 molar ratio, were solubilized in DI water and dropwise added into the alginate solution. The pH was controlled using the HCl solution at a value of 5. The solution was stirred for 2 h at RT, and then the pH was raised to 8.5 by slowly adding the corresponding amount of 0.5 M NaOH solution. PA was subsequently added directly into the flask. The reaction was left in the dark under stirring at RT for 16 h. Using this general procedure, two different molar ratios for the reactants were assayed: SA/EDC/NHS/PA 1:1:1:1 (SA–PA-1) and 1:4:4:4 (SA–PA-4). At the end of the reaction, the color of the solution turned pale yellow. The product (SA–PA) was then dialyzed for 4–6 days in DI water at RT in the dark with a 3.5 KDa membrane. The degree of functionalization was determined via NMR: ^1^H-NMR, and ^13^C-NMR spectra were recorded on a Bruker Avance 400 MHz and Varian 500 MHz spectrometers with samples dissolved in D_2_O at room temperature. The presence of the yne moieties on the alginate backbone was further confirmed by both ATR-FTIR spectroscopy, conducted using a FTIR PerkinElmer Spectrum One spectrometer, and Raman spectroscopy, conducted using a Renishaw inVia Reflex system fitted with a 515 nm solid-state laser and a CCD detector coupled to a confocal microscope. The Raman spectra were processed using Renishaw WIRE 3.4 software.

### 2.3. Hydrogel Preparation via Photo-Crosslinking

A 5% wt solution was prepared by dissolving 100 mg of SA–PA-4 in 1.9 mL of DI water. Then, 2 mg of LAP previously dissolved in 0.1 mL of DI water (2% wt with respect to SA–PA-4) was added, and all reactants were stirred together in the dark until complete solution.

Different amounts of crosslinkers (DTT or PEG-SH) were solubilized in DI water until a homogeneous solution was evidenced. Then, the crosslinkers were added to the SA–PA-4 solution and stirred in the dark until the complete homogenization of the products was achieved. The formulation was poured into PDMS molds (≈H = 3 mm, D = 5 mm) and irradiated for 5 min with polychromatic visible light from a Hg-Xe lamp (Hamamatsu LC8 Lightningcure^TM^) fitted with a cut-off filter for λ < 400 nm and a light guide (50 mW/cm^2^).

### 2.4. Hydrogel Characterization

Real-time photorheological measurements were performed using an Anton PAAR Modular Compact Rheometer (Physica MCR 302, Graz, Austria) in parallel-plate mode (25 mm diameter), and the visible-light source was provided by positioning the light guide of the visible Hamamatsu LC8 lamp under the bottom plate. During the measurements, the gap between the two glass plates was set to 0.2 mm, and the sample was kept under a constant shear frequency of 1 Hz. The irradiating light was switched on after 60 s to allow the system to stabilize before the onset of polymerization. According to the preliminary amplitude sweep measurements, all the tests were carried out in the linear viscoelastic region at a strain amplitude of 50%. The photo-rheology was studied as a function of the changes in the shear modulus (G′) and loss modulus (G″) of the sample versus the exposure time.

Amplitude sweep tests were performed on the cured hydrogels in the range of 1 to 1000% strain, with a frequency of 1 Hz.

The different photocured samples (≈h = 3 mm, d = 5 mm) were washed and left to dry overnight. Once dry, the samples were weighed and soaked in DI water to evaluate the swelling capability and kinetics. The samples were taken out at different time intervals and weighed once the surface droplets were wiped off with wet paper until a constant weight was reached. The swelling ratio (Sw%) was calculated as:(1)$$\mathrm{Sw}\;\left(\%\right)=\frac{W_{t}\;-{\;W}_{0}}{W_{0}}\;*\;100$$
where W_t_ is the weight of the hydrogel sample at a specific time, and W_0_ is the initial weight of the dried sample. All tests were performed in triplicate.

To determine the gel content (GC), previously dried samples were held in a metal net, weighed, and then immersed in DI water (25 °C) for 24 h to dissolve the un-crosslinked polymer. The samples were then dried for 24 h (40 °C) in a vacuum oven and weighed again. The gel content was determined as:(2)$$\mathrm{GC}\;\left(\%\right)=\frac{W_{i}}{W_{f}}*100$$
where W_i_ is the initial weight and W_f_ is the weight after extraction.

The morphological characterization of the samples was carried out by field emission scanning electron microscopy (FESEM, Zeiss Supra 40, Oberkochen, Germany). The hydrogel samples were first frozen, sectioned in half, and lyophilized before coating with a 5 nm thick thin film of Pt/Pd.

Mechanical properties were evaluated by a dynamic compression test. Measurements were performed on photocured samples (≈h = 9 mm, d = 5 mm) at RT using a universal test system, MTS QTest1/L Elite, a uniaxial testing machine equipped with a 100 N load cell in compression mode. Samples were placed between the compression plates. Each sample was subsequently deformed at 1 mm/min. All measurements were performed in triplicate.

### 2.5. Cell Viability and Proliferation

Before the cell viability and proliferation assays, all the hydrogels were sterilized in a 48-well plate (Corning, Corning, NY, USA). The hydrogels were stored in 70% ethanol for a week; carefully rinsed with PBS (phosphate-buffered saline, Thermo Fisher, Waltham, MA, USA); and then sterilized with ultraviolet germicidal irradiation (UVGI) for 40 min. After a final rinse with PBS, the hydrogels were covered with DMEM 1X (Gibco) supplemented with 10% FBS (fetal bovine serum, Thermo Scientific) and antibiotics (100 U mL^−1^ penicillin and 100 μgmL streptomycin sulfate (Sigma-Aldrich, St. Louis, MI, USA)). After 24 h of contact between the culture medium and the hydrogels at 37 °C, the media containing soluble extracts were collected and kept in the freezer until further use.

Cell assays were performed using C166-GFP mouse endothelial cell line (ATCC CRL-2583™, (ATCC, Manassas, Virginia USA): 20,000 cells/mL were seeded in a 24-well culture plate and allowed to adhere and grow for 24 h. Then, the media were changed for mixtures (1:1 and 1:5) of complete DMEM and the medium that had been in contact with the hydrogels.

Inverted fluorescence microscopy (Olympus IX51, FITC filter λex/λem = 490/525 nm) was used daily to evaluate any changes in the cell culture morphology and proliferation that could indicate the leaching of toxins from the hydrogels. After 48 h, when the cell cultures reached confluency, the metabolic activity of the cells was measured using an Alamar Blue assay, following the instructions of the manufacturer (Biosource). This method is non-toxic and uses the natural reducing power of living cells, generating a quantitative measure of cell viability and cytotoxicity. Briefly, Alamar Blue dye (10% of the culture volume) was added to each well containing living cells and incubated for 90 min. Then, the fluorescence of each well was measured using a Synergy HT plate reader (BioTek, Winooski, VT, USA) at 535/590 nm.

Finally, the DNA quantitation of cells was determined by fluorescent staining with a FluoReporter^®^ Blue Fluorometric dsDNA Quantitation Kit. This method is based on the ability of the bisbenzimidazole derivative Hoechst 33258 to bind to A-T-rich regions of double-stranded DNA. After binding to DNA, Hoechst 33258 exhibits an increase in fluorescence, which is measured at the 360 nm excitation wavelength and 460 nm emission wavelength using a microplate reader (BioTek, Synergy HT).

### 2.6. Statistical Analysis

An unpaired Student’s *t*-test (GraphPad Prism4) was performed to compare the metabolic activity and DNA content values of each sample. A *p*-value of less than 0.1 was considered statistically nonsignificant.

## 3. Results and Discussion

### 3.1. Synthesis of SA–PA

Amidation reactions in sodium alginate employing the coupling mechanism of EDC/NHS have already been reported in numerous investigations with different degrees of modifications [36,37,38]; the choice of alginate in these reactions is usually driven by the high amount of carboxylic acid present in the backbone. As far as we know, there no precedent in the literature for the thiol–yne functionalization of alginate as it is described here. Herein, we investigated the yne functionalization of sodium alginate by varying the ratio between the functionalization molecules (coupling agents and propargylamine) in a two-step reaction. The reaction scheme is illustrated in Figure 1.

First, the polysaccharide was solubilized in DI water, lowering the pH to 4. This pH value was important at this step to assure that all the alginate carboxylic groups were fully protonated [39]. The subsequent attack on the -COOH groups by EDC and the formation of a stable activated ester in the same site by NHS prepared the polysaccharide for further amidation with PA [40,41,42]. Equally, the effective NHS-ester attack by amines could only take place if they were in a neutral state, which was not possible at an acidic or physiological pH [43]; for this reason, before the addition of PA, the pH was increased to 8.5 to enhance the amine nucleophilicity [44,45]. Once the optimal pH was reached, the corresponding amount of PA (considering a one-to-one reaction between PA and EDC/NHS) was added to the reaction, which was then stirred in the dark to allow its completion. When an equimolar ratio between the coupling agents and the alginate carboxylic acids was employed (SA–PA-1), a very small degree of functionalization was achieved (below 2–3%), independently of the pH values chosen and the duration of the reaction. Finally, when employing a four-fold molar ratio in respect to the carboxylic alginate groups (SA–PA-4), the successful amidation of the alginate took place.

Due to the position of the alginate signals in the ^1^H-NMR spectra, the signal corresponding to the triple bond fell between those of the alginate skeleton and could not be precisely integrated (see Appendix A). A more reliable signal to quantify the number of triple bonds in the alginate structure was provided by the methylene group of PA. Its presence and position were first unambiguously identified by a heteronuclear single quantum coherence experiment (HSQC, Figure 2A), which was able to determine the carbon-proton single-bond correlation. In the figure, the -CH and -CH_3_ groups are indicated in red, whereas -CH_2_ is indicated in blue. As highlighted in Figure 2A, the signal at 3.35 ppm in the ^1^H-NMR spectrum correlated with the carbon at 35 ppm in the ^13^C-NMR spectrum, which corresponded to the PA methylene group [44,46]. The yne quantification of the SA–PA products was estimated by the integration of the 3.35 ppm signal, considering the anomeric -CH- of alginate at 5.05 ppm as an internal standard (Figure 2B). Despite this, the integration of the peaks at 3.35 ppm and 5.05 ppm was very difficult in the SA–PA-1 sample, and the degree of functionalization could only be estimated to be lower than 2–3%. The ratio between the integrals of both signals in the SA–PA-4 sample allowed an estimation of around 28% for the degree of functionalization. For this reason, SA–PA-4 was chosen as the starting material for further characterizations.

The presence of the yne moieties in SA–PA-4 was also investigated through ATR FTIR and Raman spectroscopy. While FTIR spectroscopy relies on the absorption or transmission of light with a wide range of wavenumbers, Raman spectroscopy involves the study of inelastic scattering from lamps with specific wavenumbers. These techniques are usually complementary, as some chemical bonds are more active and visible in Raman spectroscopy, especially if the proportion between bonds is low [47]. Alkynes usually fall into this category, and that is why a sharp and well-defined peak could be noted on the Raman spectrum shown in Figure 3A at 2122 cm^−1^ [48]. For the sake of completeness, Raman spectroscopy was also compared with FTIR spectroscopy for the absorption spectra of synthetized SA–PA-4 and PA (Figure 3B); low light absorption could be seen in SA–PA-4 at the same wavenumber as the triple bond of PA [49], again supporting the successful functionalization of the natural polymer.

### 3.2. Preparation and Characterization of Thiol–yne Hydrogels

Two dithiol crosslinkers of different chain lengths and molecular weights, both already examined in tissue engineering applications [50], were selected to prepare thiol–yne hydrogels, namely dithiothreitol (DTT, chain length = 4, Mw 155) and poly(ethylene glycol) dithiol (PEG-SH, chain length ≈ 24, average Mn 1000). According to the thiol–yne mechanism, each alkyne reacts first with a single thiol to form a vinyl sulfide; then, if a second thiol approaches the reactive species, the addition of a second thyil radical takes place, and a dithioether is formed [51]. Thus, eight different DI-water-based formulations (four based on DTT and four based on PEG-SH) with incremental amounts of crosslinkers were prepared to create hydrogels, as shown in Table 1.

An LAP visible-light photo-initiator was chosen because of its low biotoxicity [52]. The quantity of crosslinkers was chosen to modulate the rigidity of the hydrogels according to the future possible applications (a stoichiometric ratio between the yne moieties and the crosslinkers was used in the 0.3 DTT or S-PEG formulations). The actual network formation, system reactivity, and optimal irradiation time were investigated for all the formulations by photo-rheology (Figure 4A,C). After an initial stabilization time of 60 s, the lamp was switched on and the increase in the storage/loss modulus was recorded over time. The gel point of the S-PEG formulations (the timepoint at which the solution underwent gelation, represented by the crossover between G′ and G″, Table 2) was lower than that of the DTT molecules in all cases, with a minimum of 95 s for S-PEG 0.2, indicating a higher reactivity for these crosslinkers. Moreover, the sharper slope of the curve in the S-PEG formulations as well as the clear and defined upper plateau suggested an enhanced rate of polymerization when this crosslinker was used, even though the 0.1 DTT hydrogel possessed a slightly better shear storage modulus. This was not surprising, due to the shorter molecular weight of the DTT crosslinkers and the subsequent increased final mechanical rigidity [53]. However, the lower molecular weight of the DTT crosslinkers could also explain the reduced reactivity of the systems; in fact, once the molecule was one-side bonded, its mobility was significantly lower than that of high-molecular-weight crosslinkers. Instead, long-molecular-chain crosslinkers possess long, flexible chains, increasing the possibility of meeting a second alginate alkyne reactive site [54]. The best properties in terms of inhibition time (the time needed for the storage modulus to increase from the initial bottom plateau), polymerization rate, and final G′ value were obtained, respectively, from the 0.1 DTT and 0.2 S-PEG hydrogels. All the experimental data are reported in Table 2. The amplitude-sweep measurements (Figure 4B,D) supported the aforementioned results, with the lowest yield points (the strain value at which the hydrogel started to collapse (Table 2)) obtained for the 0.1 DTT and 0.2 S-PEG hydrogels. Indeed, a more crosslinked structure was proposed for 0.2 S-PEG, while a shorter length between crosslinks was suggested for 0.1 DTT; this theory could also explain the higher fragility of the 0.1 DTT hydrogel, as measured by the lower yield point [55].

Considering the degree of the substitution of the SA–PA-4 polysaccharide (28%) and the thiol–yne mechanism, the best properties overall were expected from the 0.3 DTT and 0.3 S-PEG hydrogels, respectively. However, while the worst properties at a lower molar ratio between the functionalized alginate and the crosslinkers (0.05 DTT, 0.05 S-PEG, and 0.1 S-PEG) were caused by a non-complete network formation (as confirmed by the low value of gel content, Table 2), the slightly worse properties of the 0.2 DTT, 0.3 DTT, and 0.3 S-PEG formulations could be explained by two different mechanisms.

The first mechanism involves the radical-scavenging properties of thiols in water environments (such as the human body). For example, many studies have proven the radical-scavenging behavior under physiological conditions of glutathione (GHS) or thiols in general, which is mostly related to the beneficial effects of preventing biological free radicals [56,57,58,59]. One of the radical-scavenging paths (the most common) follows the reaction:R• + SH ⇋ RH + S•(3)
S• + S−⇋ SS•−(4)
SS•− + O2 → SS + O2•−(5)

This reaction can be kinetically driven in the direction of removing thiyl radicals through a rapid reaction with thiolate anions (which are always present at a physiological pH in water, contrary to bulk polymerization). If the SS•− product meets oxygen, the reaction becomes irreversible at a near-diffusion-controlled rate [60]. Moreover, it was proven than dithiols, especially DTT, are more effective as radical scavengers [61,62]. In fact, after photopolymerization, especially with DTT crosslinkers, the hydrogels were not optically transparent but mildly turbid, a symptom of disulfide formation [63]. To illustrate the reaction between thiols, Figure 5 depicts the transparent appearance of the hydrogels after washing, compared to their opaque appearance just after photopolymerization. This observation could support the idea of disulfide creation between the crosslinkers and the subsequent washing away of the unbounded molecules after swelling, which is also suggested by the low gel content observed in all the formed hydrogels (Table 2).

Apart from the radical-scavenging effect of thiols in water environments, thiols are also known to act as chain transfer agents in free-radical polymerization [64,65,66]. Especially in aqueous solutions [67], a higher thiol concentration could lead to incremental radical chain transfer phenomena and a subsequent decrease in crosslinking, explaining the poorer mechanical properties at higher molar ratios.

Moreover, even if thiol–yne reactions exhibit a one yne/two thiols stoichiometric ratio, a certain degree of homopolymerization between the alkynes and vinyl sulfides is often exhibited [51].

The three mechanisms combined could explain the decreased photo-rheological properties in thiol–yne stoichiometric conditions.

The swelling kinetics of the different hydrogels were also evaluated. The different formulations were irradiated for 5 min (50 mW/cm^2^) in cylindrical molds (H ≈ 3 mm, D ≈ 5 mm), leading to the formation of hydrogels presenting differences in mechanical resistance. The cylindrical hydrogels were soaked in DI water and weighted at different time points (being careful to remove the extra water by wiping). Figure 5A,B report the swelling kinetics and capability of the studied hydrogels, while Table 2 reports the swelling equilibrium and time to equilibrium values. The same trend was observed as in the photo-rheology properties, with 0.1 DTT and 0.2 S-PEG presenting the lowest level of swelling in their categories, supporting the proposed mechanism. Between the two best hydrogels in each category, similar values were observed (781 and 791% swelling for 0.1 DTT and 0.2 S-PEG, respectively). Interesting, in all cases the obtained swelling equilibriums were comparable to or higher than the corresponding values reported for (meth)acrylate photocured alginate hydrogels [68,69,70,71]. For all these reasons, 0.1 DTT and 0.2 S-PEG were selected for further investigations.

The 0.1 DTT and 0.2 S-PEG samples were freeze-dried, and their inner architecture was observed by field emission scanning electron microscopy (FESEM). Both samples presented the typical porous structure that is required in hydrogels designed for scaffold/filler applications. As is visible in Figure 6A,B, the 0.1 DTT sample presented a less homogeneous structure, while the 0.2 S-PEG sample showed a more compact network with regular porosity.

The two selected hydrogels were also subjected to a compression test in a cylindrical shape (Figure 6C, d = 5, h = 9). The results showed mechanical properties in the same order of magnitude for the 0.2 S-PEG and 0.1 DTT hydrogels in terms of storage modulus (E′), ultimate compression strength, and deformation at break (See Figure 6C). Moreover, the absolute mechanical values reported here were comparable to or higher than those of other proposed alginate hydrogels obtained using methacrylates [72,73] or thiol–ene reactions [36]. The better values obtained for 0.2 S-PEG with respect to 0.1 DTT could be attributed to the more homogeneous network, as was also observed in the FESEM images, and to the more deformable crosslinked structure (due to the higher molecular weight of the PEG crosslinker).

### 3.3. Cell Viability and Proliferation

The different hydrogels were sterilized as described above and immersed in complete medium for 24 h at 37 °C previous to the biological evaluation. Laterally, autofluorescent C166-GFP endothelial cells were seeded and allowed to adhere for 24 h in a 12-well plate. According to ISO 10993-5 recommendations, to ensure that no toxic substances were released from the hydrogels, culture media that had been in contact with the hydrogels were added to the endothelial cells growing in the culture plate. Their proliferation was assessed via inverted bright-field microscopy for 48 h (Figure 7A). At this point, their metabolic activity and DNA content was quantified to assess hydrogel cytocompatibility (Figure 7B,C, respectively).

As can be seen in the micrographs (Figure 7A), no morphological changes were observed after culture media replacement. No detached cells or evidence of necrotic or apoptotic cell bodies were identified. In all samples, a healthy and confluent cell monolayer was photographed. With respect to the mitochondrial metabolic analysis (Alamar Blue assay, Figure 7B), all samples showed proper levels of cell activity. Accordingly, the DNA proliferation analysis (Figure 7C) corroborated our previous observations, with similar levels of DNA content for all conditions. Additionally, no statistically significant differences were found, either in the metabolic activity or in the DNA content of the cultured cells when compared to a control in which the media were not changed.

In summary, no thiol–yne alginate hydrogel showed any signs of indirect in vitro cytotoxicity. These results open up new possibilities for tissue engineering applications of these hydrogels in clinical therapies. According to our results and considering the presented mechanical properties, DTT alginate hydrogels (0.1 DTT) could be implemented in clinical indications with no strict structural requirements, e.g., as protective dermal patches or other wound dressings [74]. In contrast, based on their mechanical properties, the 0.2 S-PEG hydrogels could be used as surgical fillers in cosmetic plastic surgery procedures or after tumour resections. In these situations, alginate hydrogel fillers are commonly used to maintain body structures or to sustain tissue proliferation and organism regeneration [75].

## 4. Conclusions

The successful and simple production of alginate hydrogels by thiol–yne reactions was here reported for the first time; in all cases, the properties of the hydrogels were suitable for wound-dressing or surgical filler applications. Well-known carbodiimide chemistry methods allowed us to achieve up to 28% substitution of the yne moieties on the alginate backbone, permitting the formation of hydrogels using two different types of dithiol crosslinkers, i.e., 1,4-dithiothreitol (DTT) and poly(ethylene glycol) dithiol (PEG-SH). The gel characterization showed the reactivity of the system within a few minutes of irradiation, despite the radical-scavenging and chain transfer properties of thiols. Furthermore, the mechanical, rheological, topographical, and swelling properties of the formed gels presented similar or higher values compared to the standard and more cytotoxic (meth)acrylate photocured alginate hydrogels, supporting the proposed applications of these materials. Lastly, the hydrogels obtained by thiol–yne reactions showed no signs of releasing cytotoxic byproducts that would impede their biomedical applicability. Cells in contact with culture media extracts exhibited optimal proliferation measured in terms of metabolic activity and DNA content. In conclusion, these new cytocompatible hydrogels are promising candidates for medical applications such as wound dressings or surgical fillings.

## Figures and Tables

**Figure 1 polymers-14-04709-f001:**
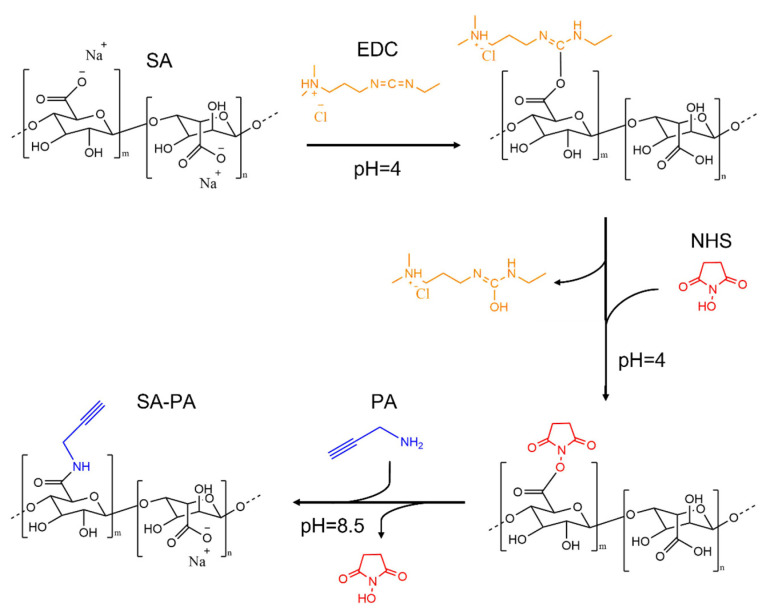
Reaction scheme of the two-step thiol–yne functionalization of alginate (SA–PA) by carbodiimide chemistry.

**Figure 2 polymers-14-04709-f002:**
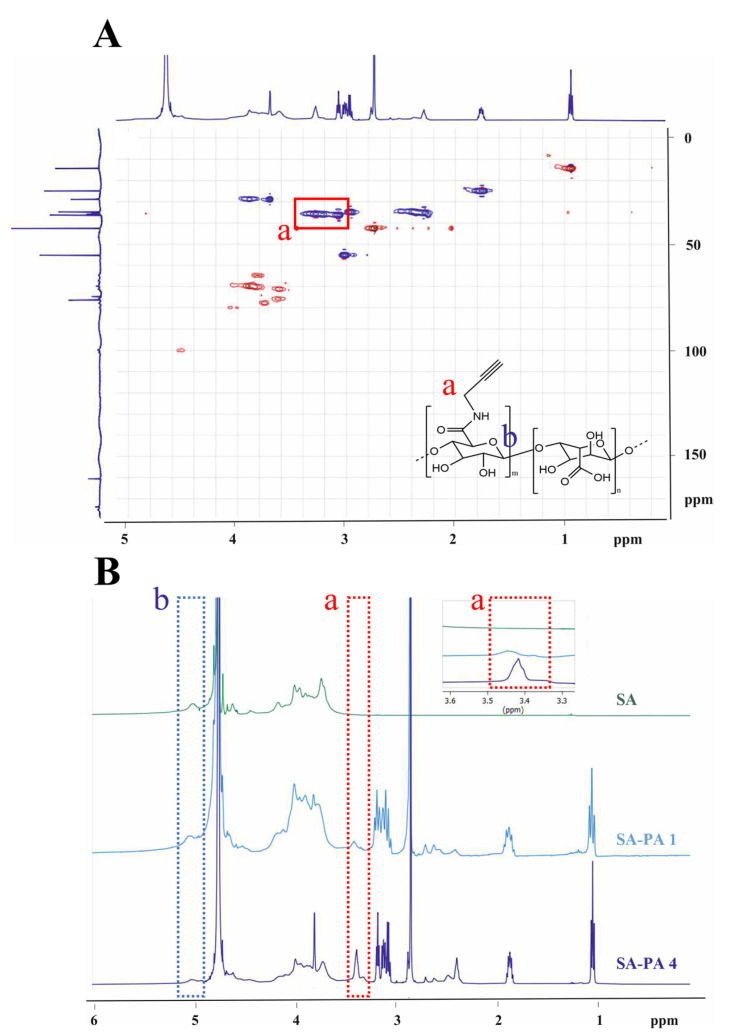
HSQC (**A**) and ^1^H-NMR (**B**) spectra of SA–PA products. The PA methylene is indicated as (a), while the anomeric methyl of alginate is indicated as (b).

**Figure 3 polymers-14-04709-f003:**
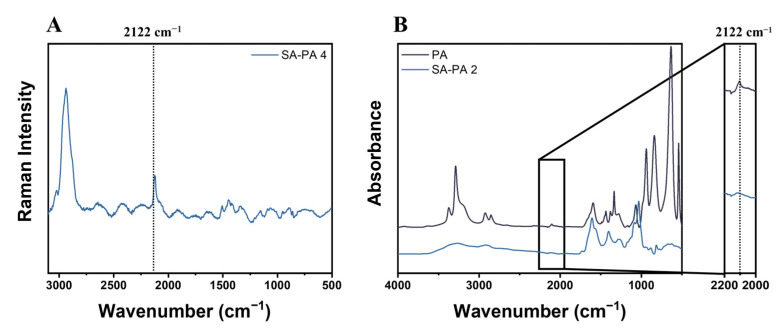
Raman spectroscopy of SA–PA-4 with the alkyne bond highlighted at 2122 cm^−1^ (**A**) and infrared spectra of SA–PA-4 and PA (**B**) with the same bond evidenced.

**Figure 4 polymers-14-04709-f004:**
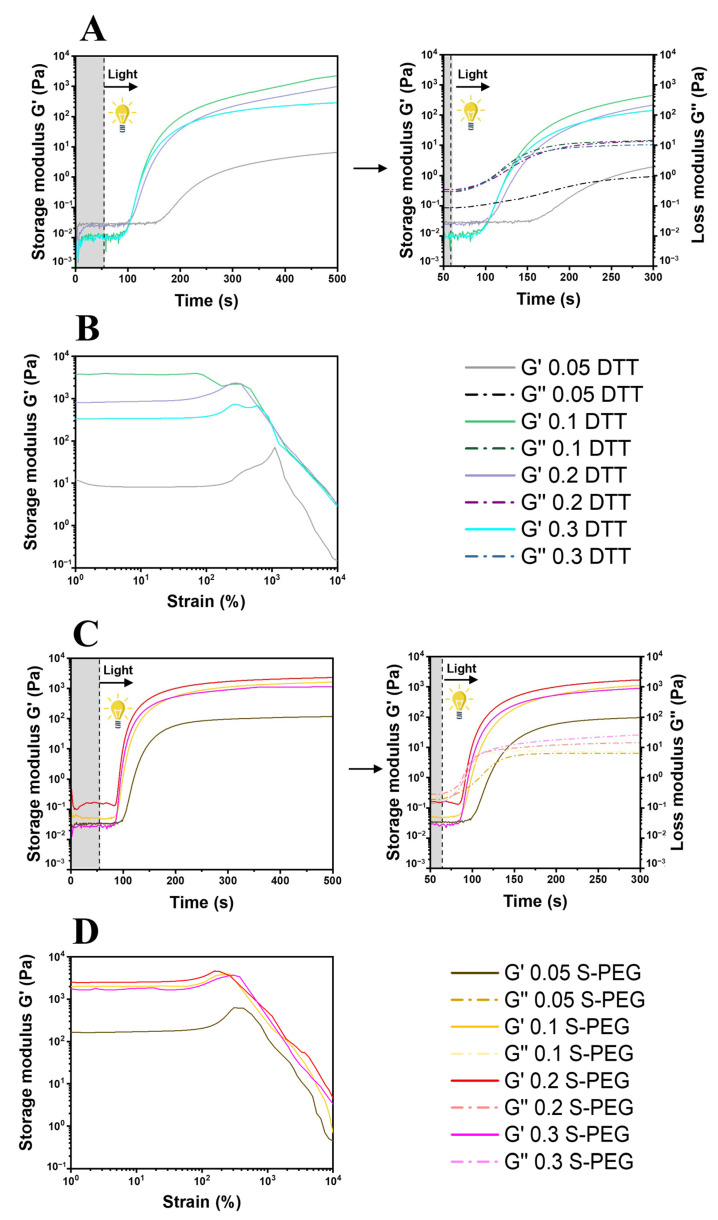
Photo-rheology of the DTT/PEG systems ((**A**)/(**C**), respectively) and amplitude sweep of the same systems ((**B**)/(**D**), respectively).

**Figure 5 polymers-14-04709-f005:**
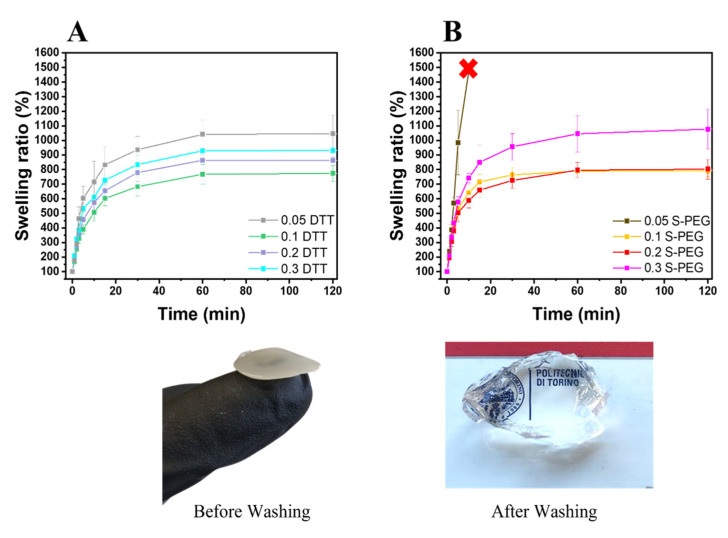
Swelling kinetics of the DTT (**A**) and PEG hydrogels (**B**). In the images below the graphs, the 0.2 S-PEG hydrogel is depicted before and after washing.

**Figure 6 polymers-14-04709-f006:**
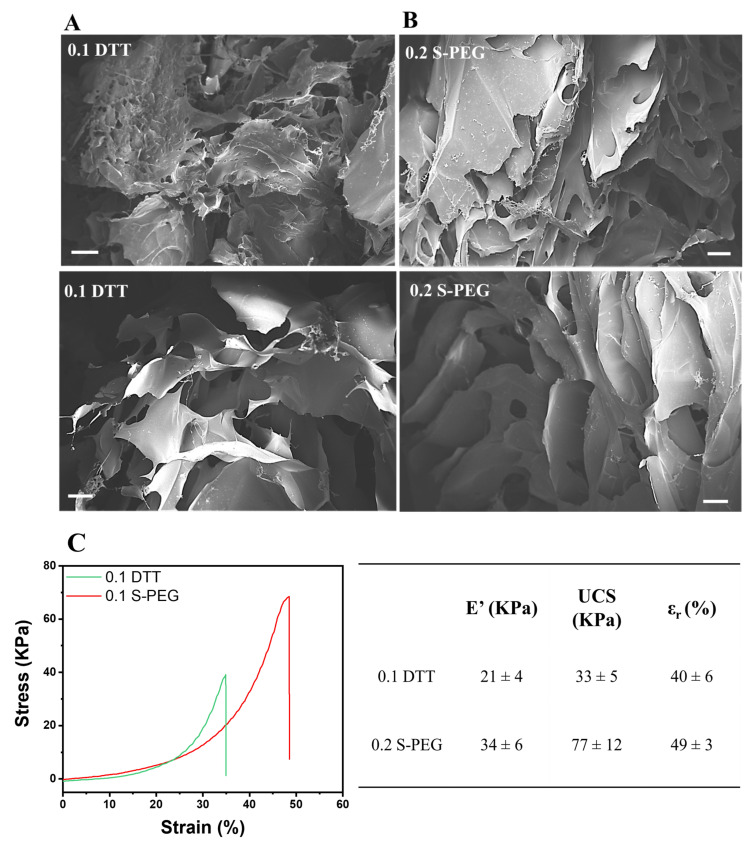
Field emission scanning electron microscopy morphology of 0.1 DTT (**A**) and 0.2 S-PEG (**B**) hydrogels; scale bar 200 µm. (**C**) Hydrogel compression test using the two different crosslinkers; the table on the right shows the reported values of compression elastic modulus (E′), the ultimate compression strength (UCS), and the elongation at rupture (ε_r_). All values are reported with standard deviation.

**Figure 7 polymers-14-04709-f007:**
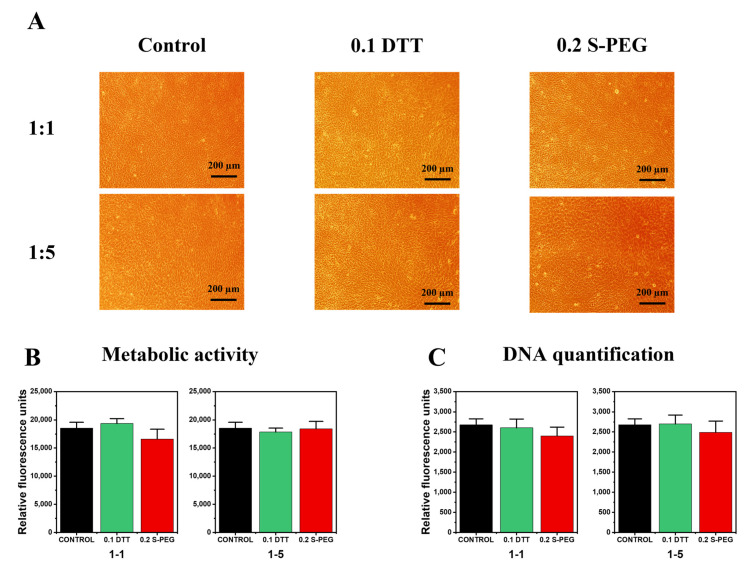
Bright-field microscopy (**A**), metabolic activity (**B**), and DNA quantification (**C**) of the two hydrogels 0.1 DTT and 0.2 S-PEG.

**Table 1 polymers-14-04709-t001:** Thiol–yne formulations. All the hydrogels included 5 wt% SA–PA-4 and 2 phr of LAP photo-initiator and were irradiated for 5 min.

Sample	Crosslinker (HS-R-SH)	Molar Ratio (SA–PA:HS-R-SH)
0.05 DTT	DTT	1:0.05
0.1 DTT	DTT	1:0.1
0.2 DTT	DTT	1:0.2
0.3 DTT	DTT	1:0.3
0.05 S-PEG	PEG-SH	1:0.05
0.1 S-PEG	PEG-SH	1:0.1
0.2 S-PEG	PEG-SH	1:0.2
0.3 S-PEG	PEG-SH	1:0.3

**Table 2 polymers-14-04709-t002:** Rheological, mechanical, and swelling properties of the thiol–yne hydrogels.

Sample	Gel Point (s)	Induction Point (s)	Time to Plateau (s)	Storage Modulus G′ (KPa)	Yield Point (%)	Swelling Equilibrium (%)	Swelling Time to Plateau (h)	%GEL (%)
0.05 DTT	237	153	/	0.03	1110	1045 ± 26	1	41 ± 6
0.1 DTT	139	87	/	3.8	89	781 ± 63	1	72 ± 1
0.2 DTT	153	97	/	0.81	337	835 ± 46	1	62 ± 0.5
0.3 DTT	142	87	/	0.3	613	932 ± 16	1	53 ± 4
0.05 S-PEG	129	100	≈200	0.11	570	/	1	33 ± 4
0.1 S-PEG	100	86	≈200	1.7	185	802 ± 37	1	70 ± 3
0.2 S-PEG	95	86	≈200	2.3	266	791 ± 47	1	75 ± 2
0.3 S-PEG	98	86	≈200	1.1	377	1306 ± 222	1	57 ± 4

## Data Availability

Not applicable.

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
