# Peer review of "Photocurable Thiol–yne Alginate Hydrogels for Regenerative Medicine Purposes"

_polymers, 2022, doi:10.3390/polym14214709_

Round 1

Reviewer 1 Report

This is a well-done study about photocurable alginate hydrogels for regenerative medicine. I recommend it for publication after the minor points below are addressed.

1. In the section of Materials, the Mn of the PEG is equal to 1000, while Mw of the PEG is roughly equal to 1000 in the section of Results and Discussion. The authors should specify the PDI of the PEG. 

2. The authors claimed that mechanical properties were tested. However, no compression and extension tests were done (only rheological tests).

3. Several recent studies about hydrogels (Acta Biomaterialia 140, 324-337, 2022; Science 370 (6514), 335-338, 2020) used in regenerative medicine should be included in the section of the Introduction.

4. formatting issues. Line 83, 'Alginic acid sodium alginate' to 'Sodium alginate'; title ‘Photocurable Thiol-Yne Alginate Hydrogels for regenerative medicine purposes’ to ‘Photocurable Thiol-Yne Alginate Hydrogels for Regenerative Medicine Purposes’. Please check all.

Author Response

We thank the reviewer for the kind evaluation of the work and for the useful comments. We did our best to fulfill the reviewer’s requests. Here below the point by point response to the comments.

  1. In the section of Materials, the Mn of the PEG is equal to 1000, while Mw of the PEG is roughly equal to 1000 in the section of Results and Discussion. The authors should specify the PDI of the PEG. 

Thank you for the clarification. We wrongly reported the indication of rough Mw in the Results and Discussion part. The material used was purchased by Merck Sigma-Aldrich; the right indication reported on the datasheet is “average Mn 1000”as now reported in the paper.

  1. The authors claimed that mechanical properties were tested. However, no compression and extension tests were done (only rheological tests).

Thank you for the suggestion. Compression tests were now performed and added to the discussion.

  1. Several recent studies about hydrogels (Acta Biomaterialia 140, 324-337, 2022; Science 370 (6514), 335-338, 2020) used in regenerative medicine should be included in the section of the Introduction.

Thank you, we included the very interesting references you suggested.

  1. formatting issues. Line 83, 'Alginic acid sodium alginate' to 'Sodium alginate'; title ‘Photocurable Thiol-Yne Alginate Hydrogels for regenerative medicine purposes’ to ‘Photocurable Thiol-Yne Alginate Hydrogels for Regenerative Medicine Purposes’. Please check all.

Thank you, we changed the sentences

Reviewer 2 Report

This manuscript mainly described the first alginate hydrogel obtained by thiol-alkyne reaction, which has good cytocompatibility.

In my opinion, this article is complete in structure and clear in organization. But some discussions may be ambiguous and corresponding explanations and experiments need to be supplemented. What's more, this manuscript is rich in content and has a certain reference value in bio-based hydrogel. Therefore, I suggest that this article can be published, but the paper needs suitable revision before acceptance for publication. My detailed comments are as follows:

1. In this paper, the successful preparation of SA-PA needs to provide more characterization, complement the mass spectrum or FTIR, and integrate the NMR hydrogen spectrum.

2. The authors should try to give more data characterization for hydrogels preparation: such as XPS, and element analysis or other performances need to be supplemented.

3. Please further elaborate on the advantages of using the thiol-alkynene method to prepare hydrogels, such as increasing performance comparison with other hydrogels.

4. The authors could add the following references which would again increase the interest to general functional material prepared by thiol-based click reactions: Journal of Controlled Release‚ 2018‚ 273, 160-179; Polymer‚ 2017, 125, 303-329; Chemical Society Reviews‚ 2022‚ 51, 4175-4198.

Author Response

Reviewer 2

We thank the reviewer for the evaluation of the work and for the useful comments. We did our best to fulfill the reviewer’s requests, here below our answers to the comments.

  1. In this paper, the successful preparation of SA-PA needs to provide more characterization, complement the mass spectrum or FTIR, and integrate the NMR hydrogen spectrum.

Thank you for the suggestion, FTIR spectrum is compared to the RAMAN Spectrum in Figure 3; while the H-NMR spectrum integration has now been reported in the Supporting Info File for Alginate and samples SA-PA 1 and SA-PA 4. All the modifications reported are now highlighted in the manuscript.

  1. The authors should try to give more data characterization for hydrogels preparation: such as XPS, and element analysis or other performances need to be supplemented.

We agreed with the Reviewer that the characterization could be improved, so, also according to Reviewer1’s requests, we now performed compression tests and implemented the discussion in the paper. In such a short time, we chosed to deepen the characterization of the final performances since this better fits with the final aim of the work.  

  1. Please further elaborate on the advantages of using the thiol-alkynene method to prepare hydrogels, such as increasing performance comparison with other hydrogels.

Thank you for your suggestion. The main reason to use this kind of reaction to produce soft hydrogels is related to the lower cytotoxicity of the yne/thiol moieties compared to (meth)acrylates, as pointed out in the introduction (line 63). To better contextualize the advantages and performances of the proposed hydrogels  we now included new sentences and references to make the readers appreciate the mechanical behavior of the hydrogels.

  1. The authors could add the following references which would again increase the interest to general functional material prepared by thiol-based click reactions: Journal of Controlled Release‚ 2018‚273, 160-179; Polymer‚ 2017, 125, 303-329; Chemical Society Reviews‚2022‚ 51, 4175-4198.

Thank you, we included the very interesting references you suggested.